# OpenReview forum: "What Makes Your Model a Low-empathy or Warmth Person: Exploring the Origins of Personality in LLMs"
_ICLR.cc/2025/Conference — ICLR 2025 Conference Withdrawn Submission_

### Official Review · Reviewer_PxVq · 2024-10-29

**Soundness:** 1
**Presentation:** 2
**Contribution:** 1
**Rating:** 1
**Confidence:** 5

**Summary:**

The paper attempts to investigate how long-term factors, represented as activation features of LLMs, and short-term pressures, operationalized by prompt engineering, affect the LLMs' ability to complete the Big Five Inventory and Short Dark Triad personality tests. The authors show that the models exhibit less agreeableness and more neuroticism on the Big Five Inventory test when they have been tuned to have strained family relationships. They also show that if the models are prompted to be gregarious through prompts like 'Imagine you're a person who enjoys being around others and thrives in social situations', the models are more likely to score lower on Psychopathy and Machiavellianism on the Dark Triad personality test.

**Strengths:**

The authors have robust methods and solutions. The main strength of the paper is the range of factors the authors tested. These include gender, age, education, social ideology, etc. The results are presented in an extremely accessible manner in the tables.

**Weaknesses:**

While this paper has many issues, the main one is the framing. The authors are attempting to explore personality in LLMs. There seems to be a fundamental misunderstanding of what personality entails. Personality definitionally presumes personhood (https://www.apa.org/topics/personality). As the authors point out in the Sparse Autoencoders (SAEs) Section, they are investigating activations in LLMs, not personality.








This misunderstanding of what personality entails is further evidenced by the works they have cited, e.g., Joshi et al. (2023a). The cited paper discusses personas, not personalities. Personas and personalities are not interchangeable ideas. Personality refers to the intrinsic qualities and characteristics that make an individual who they are, while persona refers to the outward projection.




The LLM's ability to solve the Big Five Personality MCQs or the Dark Triad test does not indicate that the LLM itself has a personality; it just indicates that it can solve the tests. The authors might be conflating the ability to process and respond to test questions with having the underlying psychological constructs being measured.




The authors' definition of personality is extremely atypical from a cognitive psychological perspective. I would have preferred that in the background section they had gone into detail about how they are thinking of personality and how their definition compares to the typical ideas of personalities.

I would encourage the authors to engage with literature from cognitive psychology which explicitly define personality as a fundamentally biological feature. Consider the following examples from the field:

Mischel & Shoda define personality as emerging from the interaction between an organism's internal psychological features and its environment, requiring consciousness and biological systems that LLMs fundamentally lack.

McCrae & Costa grounds personality in "basic tendencies" that are rooted in biology and require a living system capable of adaptation and response to environment.


In order to make their argument more robust, I would recommend that the authors should:
1. Clarify if they are using "personality" as an analogy or metaphor rather than claiming LLMs have true personalities
2. Consider alternative frameworks like "behavioral patterns" or "response tendencies" that don't carry the same biological/psychological implications
3. Discuss the limitations of applying human personality constructs to computational systems


The authors seem to have cited works without actually engaging with them. I have pointed out the Joshi et al. (2023a) paper earlier. They also have cited Perez et al. 2023 in support of reliability concerns in LLMs, including misinformation and privacy risks. However, the cited paper does not discuss either of these issues. The paper instead cites Carlini et al. (2019, 2021) for privacy risks and Lin et al. for misinformation. I would recommend that the authors should engage with those works directly instead of Perez et al. 2023
I would also suggest that the authors review all their citations to ensure they directly support the claims made.

In line 051, the authors matter-of-factly allude to previous works that have identified two primary strategies for endowing LLMs with personality traits. However, this is unsubstantiated, as no supporting citations are provided for this claim.
I would suggest that the authors either provide specific citations for the claim about the two primary strategies. In case these claims do not exist, rephrase the statement to clarify that this is their own observation or hypothesis.

Additionally, the research questions are not sufficiently answered. In RQ2, the authors asked `how can these personalities influence LLMs' safety?’ Within the main text of the paper, this RQ was not engaged with in any form. I would recommend that the authors either consider removing or revising RQ2 to better align with the actual content of their paper. If they feel RQ2 is necessary for their argument, then they can include a dedicated section to address RQ2, providing specific analyses and results related to LLM safety.


Works Cited:

Mischel, Walter, and Yuichi Shoda. "A cognitive-affective system theory of personality: reconceptualizing situations, dispositions, dynamics, and invariance in personality structure." Psychological review 102.2 (1995): 246.

McCrae, Robert R., and Paul T. Costa. "Empirical and theoretical status of the five-factor model of personality traits." The SAGE handbook of personality theory and assessment 1 (2008): 273-294.

**Questions:**

The authors' definition of personality is extremely atypical. I would suggest the authors add a background section to discuss their definition of personality and how it compares to the cognitive psychology literature. I would have preferred that in the background section they had gone into detail about how they are thinking of personality and how their definition compares to the typical ideas of personalities.

Minor concern: The use of the phrase 'trustworthy concerns' in line 32 is quite uncommon. I am not aware of any paper using this particular phrasing. If this is a phrase introduced by the authors, they should clarify it. If this is a common phrase any references would be helpful for the uninitiated reader. An alternative phrasing would be 'reliability concerns'.

The authors have robust methods and solutions; however, there seems to be an issue with framing. I would suggest that the authors either reframe the scope of this work to a problem besides 'personality in LLMs' or include a researcher from psychology or at least discuss their work with some researchers from psychology.

I would also encourage the authors to restructure the paper so that they can sufficiently answer both RQs instead of one.
The labels in Table 5 might be incorrect. I would suggest that the authors fix those. Narcissism is rated as 4.3 for Gemma-2B-Instruct base
in Table 5, but is rated as 4.3 for Gemma2-9B-Instruct base in Tables 1 to 4.

In the both introduction and abstract, the authors have framed short-term pressures and long-term factors as interactional and related phenomena. However, in their experiments, they focus on the two features as independent and investigate them independently. These features might be interactional, and it would be prudent to study them in conjunction. I.e., assume a 'gregarious young person', 'gregarious old person', 'gregarious male', 'gregarious female', etc.

Generally, the word 'warmth' is used as a noun; however, the title 'what makes your model a low-empathy or warmth person' uses warmth as an adjective.

It is not clear from the write up why the authors have preferred GEMMA over other open source generative models like Llama. A justification of the choice would be helpful to the reader.

**Details Of Ethics Concerns:**

The research maps human psychological constructs (personality tests, trait theories) to LLMs without proper justification or acknowledgment of fundamental differences between human cognition and LLMs. This anthropomorphization of LLMs through personality frameworks could mislead both researchers and the public about the true nature and capabilities of LLMs.

The paper explores how to modify LLM behavior using factors like gender and socioeconomic status, failing to acknowledge that the LLM is fundamentally a set of matrix multiplications and lacks both gender and socioeconomic status. The authors risk conflating statistical patterns with genuine human attributes.


The research appears to use psychological assessment tools (Big Five Inventory, Dark Triad) without proper consideration of the validity of applying clinical/psychological assessment tools to non-human entities.


Publishing this paper in its current form is not contributing much to the academic community, as it promotes methodologically unsound practices, misapplication and misunderstanding of ideas from cognitive psychology and potentially misleading anthropomorphization of LLMs.

---

### Official Review · Reviewer_ko6j · 2024-10-31

**Soundness:** 1
**Presentation:** 1
**Contribution:** 2
**Rating:** 3
**Confidence:** 3

**Summary:**

This paper sought to measure the variability of LLM traits in response to steering towards different model personas (background factors like age, education, social ideology, etc) and pressures (i.e. trust, etc).

Feature steers are achieved in two ways.

One through extracting feature vectors via sparse encoders with SAE Lens, identifying highly activated features against GPT4o generated descriptions against each background (i.e. wealthy lineage, affluent upbringing -> for rich, and "struggling financially/etc -> for poor).  These feature steers are integrated into the residual stream for model steering.

Two, "Short-term pressure" features (basically, prompt-based steers, i.e. asking the model "imagine you are a person who is xxx" before generation) use generated GPT4o-generated prompts for each desired psychometric attribute (i.e. "competence") and representation engineering (Zou et al 2023) to capture their activation features, which are then passed through PCA to find unit steering vectors. These features are added to corresponding activations for model steering.

The effects of this steering are measured via a personality test for LLMs, TRAIT, which comprises of 8K multiple-choice questions against psychometric traits. Results of how these measures vary against different feature steers are described and shown, with the authors making the conclusion that larger LLMs are more easily shaped by "external" pressures, while smaller LLMs are more sensitive to the background factor of the persona, among other findings of similar nature.

**Strengths:**

The authors ask an interesting and novel question—how do model background descriptions lead to changes in model personality? To wit, the authors structure their experiments in an interpretable fashion, and ground their work in existing psychometric literature (and the emerging LLM psychometric literature), incorporating validated (for humans) psychometric questions into their prompts.

**Weaknesses:**

It's not immediately clear that the results for different personality steers are directly comparable with each other.

While the authors state that they "guarantee[d] the monosemanticity nature of each feature" by "verify[ying] that they remained inactive when presented with descriptions of other factors", even with this method, it's not especially clear if the degree to which each background steer may be captured by a single SAE feature vector is the same for each background tested, or that the feature activations are monosemantic to the specifically tested background attribute. For example, it could very well be the case that while the feature vector for "poor" is indeed monosemantic relative to "rich", and vice versa, the degree to which the feature vector is monosemantic to our understanding of "poor" is different to the degree to which the same is for the "rich" feature vector. This would mean that the conclusions observed on model personalities aren't so much driven by actual background factors (which is the aim of this paper) as they are by quirks of the steering vectors found.

This problem is compounded by steering coefficients—it's not especially clear that using the same steering coefficients across all concepts for these steering vectors that may have varying degrees of monosemanticity to the concepts tested is necessarily valid: for example, it may very well be the case that a coefficient of 200x towards "poor" is equivalent to a steering coefficient of 800x towards "rich".

These validity issues make it hard to take the conclusions of the paper at their face value—we can't say that, for instance, larger LLMs are more easily shaped by external pressure while LLMs are more sensitive to background factors, as the authors conclude here, or make conclusions of similar nature.

Nit: It was difficult parsing this paper. For instance, Tables 2-5 don't have any units on their measurements or any captions, so it was difficult to parse what these stated numbers mean. This paper would benefit from a round of revision.

**Questions:**

How did you validate that the GPT4o-generated sentence descriptions/words actually represent what you intended them to be?

---

### Official Review · Reviewer_haF1 · 2024-11-04

**Soundness:** 1
**Presentation:** 1
**Contribution:** 2
**Rating:** 3
**Confidence:** 4

**Summary:**

The authors explore the factors influencing personality traits in LLMs, drawing on social determinism theory and recent advances in interpretability. They classify these influences into long-term background factors, such as training data, and short-term external pressures, like prompts and instructions. Using SAE and representation-based methods, they analyze how these factors impact LLM personality through the Big Five and Short Dark Triad tests. The study compares two models: Gemma-2-9B-Instruct and Gemma-2B-Instruct, and analyze their differences in personality expression.

**Strengths:**

The paper’s motivation makes sense, grounded in the theory of social determinism, which provides a new framework for exploring LLM personality. The use of SAE and other interpretability methods indeed provides an interesting direction to understanding personality traits in LLMs.

**Weaknesses:**

However, there are several critical weaknesses that lead me to recommend rejecting this paper.

First, Table 5 appears to mislabel which results correspond to each model when cross-referenced with Tables 2–4, casting doubt on the reliability of the whole Section 5.2. The error also complicates my understanding whether the observations on long-term factors align consistently with short-term factors across different model sizes. I would strongly recommend verifying these results for accuracy.

In addition, many findings and claims seem speculative and lack sufficient experimental support. For example, the statement in line 454 (1) feels overly inferred. Rather than relying on abstract claims about model size and personality feature stability, it would be more insightful to analyze models’ responses to questions directly tied to each background factor. For instance, instead of personality tests, examining if steering on gender with SAE yields more significant changes on gender-related questions for the 9B model would provide clearer insights.

The claim that SAE is better suited for long-term influences while representation-based methods are better for short-term factors lacks empirical support. Looking solely at the definitions of long-term and short-term factors in the paper, either method could easily and reasonably apply to either factor set. This suggest for further justification and experimentation to substantiate this claim.

In light of these issues, I find the overall contribution insufficiently sound to get acceptance.

**Questions:**

N/A

---

### Official Review · Reviewer_FvtT · 2024-11-04

**Soundness:** 2
**Presentation:** 2
**Contribution:** 3
**Rating:** 5
**Confidence:** 4

**Summary:**

Authors explore controlling LLM ‘personalities’ using sparse autoencoders and representation-based methods to extract features that are correlated with long-term background factors and short-term social pressures that are posited to influence human personalities. They use these features to steer the models’ responses and test them on the TRAIT personality assessment tool based on dimensions of the BFI and SDT personality tests but designed for testing LLMs and against Safetybench safety benchmarking tool. Results demonstrate that their method can effectively steer the models’ performance.

**Strengths:**

The paper presents an original study using sparse auto-encoders to steer LLMs and successfully influence LLMs behavior on personality measures and safety benchmarks. The study is important for the field of AI interpretability as an empirical demonstration of LLM steering by directly manipulating features in the model layers. The study is significant as one of the first to attempt to steer LLM personality using SAE and representation-based methods.

**Weaknesses:**

The authors make some claims that are not clearly supported by the results: “Larger model exhibits more stable personalities and lower dark traits”, “Larger LLM is more easily shaped by external pressure, while smaller LLM is more sensitive to the background factor”, “Older and liberalism influence most on larger models while communism and uneducated in-
fluence most on smaller models’ personalities”, and “Larger models are driven by self-motivations while smaller models are shaped by self-confidence in skills.“ From what I can gather, these claims are based only on the relative change in scores based on the feature steering. Some of the changes are quite big, but no statistics are provided to quantify the difference, such as significance or effect size.

The presentation of the paper is a little confusing and repetitive. A clearer separation between the theoretical framework, methods, results, and discussion would greatly help the reader to understand the study. Importantly, safety is presented as one of the main objectives of the paper, yet the findings and discussion are relegated to the appendices.

The argument for social determinism in the study of LLM personalities is not convincing. From my reading, it appears that the choice of background factors and social pressures are based on empirical findings from social determinism but this is not well articulated in the paper. It is not immediately obvious that empirical findings from human personality psychology will apply to LLMs. Moreover, there is abundant literature that questions the notion of stable personalities in LLMs and this is not addressed in the paper at all. The idea of using educational attainment, cultural background, and political ideology with LLMs might appear contentious to some readers as LLMs are manifestly not human. Ultimately these factors are only used as categories for steering the LLMs and the psychological literature is not revisited in the findings, nor are the findings compared to the psychological literature. Thus, the validity or the necessity of couching this work in human psychological terms is questionable as the use of human psychological factors in the study of LLMs runs the risk of anthropomorphizing AI.

**Questions:**

P2 82 “ Our study employs SAEs to extract background features (e.g., educational level or cultural background) encoded during training”
P2 100 “We provide some insightable findings on how long-term background factors like age and Family Relations and external pressure like Achievement Striving can influence LLM’s
personality.”
-> Can LLM really be said to have an education level or cultural background? This smacks of anthropomorphization and just confuses the matter. There might indeed be parallels with human populations but it would be helpful to elucidate these links or it risks confusion, e.g., education attainment is related to the subject matter content; socio-cultural background is related to the cultural sources of the training data, etc.

P3 113 “Personality and Trait Theory on LLMs” -> This is a very superficial summary and does not present any extant findings. Research suggests that LLMs do not exhibit stable personalities and ought to be considered more like a cultural superposition of perspectives. This is at odds with your literature review, and ought to be addressed directly.

P4 193 4 SOCIAL DETERMINISM IN LLM PERSONALITY
-> I’m not sure what social determinism adds to the present discussion. Only superficial links to the psychological literature are made and no implications from theory or empirical findings are inferred. Unfortunately the conclusions are left to the reader when a fuller exposition might help clarify the intentions of the authors. Social determinism as a theory makes certain claims and predictions about human behavior. Is the reader supposed to infer that these apply to LLMs too? Are long-term background factors more powerful than short term pressures in explaining LLMs?

P6 310-316 “For background factors, we carefully chose 1-2 key elements from each domain in Table 1, ensuring comprehensive coverage of influential aspects. These include Family Environment (represented by Family Relations Status), Cultural and Social Norms (Social Ideology), Education (Education Level), Life and Work Experience (Professional Commitment), and Environmental Stressors (Socioeconomic Status). We also considered Biological Development factors (Gender, Age, and Emotional Intelligence) and the impact of Media and Technology (AI Familiarity). These factors were selected based on their significant impact on personality development, as supported by various studies in the field.”
-> How can an LLM be said to have any of these? There are some equivalences being drawn between LLMs and humans but these have not been rendered explicitly.

P8 427 -> What is the base score ?
P8 428 -> What should the largest difference be regarded as the most determinant? Even if the SAE is identifying monosemantic features, the interplay can hardly be said to monofactorial. As your results demonstrate, a change to one feature can have effects across personalty scores.
P8 Tables and 3 -> What do the values signify? Trait is only presented on the next page after the results of Table 2 and 3. Yet this is important to understanding the experimental setup. Results should not be presented in the experimental setup.

P9 and P10 -> It is unclear how these conclusions are being drawn on the basis of changes in the prompts. Some of the claims are speculative and should be separated from the presentation of the results. Statistical tests would help to understand what differences were significant.

---

### Note · Authors · 2024-11-12

I have read and agree with the venue's withdrawal policy on behalf of myself and my co-authors.